# Trophic Ecology of Endangered Gold-Spotted Pond Frog in Ecological Wetland Park and Rice Paddy Habitats

**DOI:** 10.3390/ani11040967

**Published:** 2021-03-31

**Authors:** Hye-Ji Oh, Kwang-Hyeon Chang, Mei-Yan Jin, Jong-Mo Suh, Ju-Duk Yoon, Kyung-Hoon Shin, Su-Gon Park, Min-Ho Chang

**Affiliations:** 1Department of Environmental Science and Engineering, Kyung Hee University, Yongin 17104, Korea; ohg2090@naver.com (H.-J.O.); chang38@khu.ac.kr (K.-H.C.); jinmeiyan8709@163.com (M.-Y.J.); 2Integrative Freshwater Ecology Group, Centre for Advanced Studies of Blanes (CEAB-CSIC), Blanes 17300, Spain; whdah9004@naver.com; 3Research Center for Endangered Species, National Institute of Ecology, Yeongyang 36531, Korea; grandblue@nie.re.kr; 4Department of Marine Sciences and Convergence Technology, Hanyang University, Ansan 15588, Korea; shinkh@hanyang.ac.kr; 5Invasive Alien Species Research Team, National Institute of Ecology, Seocheon 33657, Korea; park9663@nie.re.kr; 6Environmental Impact Assessment Team, National Institute of Ecology, Seochen 33657, Korea

**Keywords:** endangered amphibian species, wetland food web, isotopic niche space, SIBER, niche competition

## Abstract

**Simple Summary:**

Gaining information about the habitat environment and biological interactions is important for conserving gold-spotted pond frogs, which are faced with a threat of local population extinction in Korea due to artificial habitat changes. Based on stable isotope ratios, we estimated the ecological niche space (ENS) of gold-spotted pond frogs in an ecological wetland park and a rice paddy differing in habitat patch connectivity and analyzed the possibility of their ENS overlapping that of competitive and predatory frogs. Gold-spotted pond frogs showed a broader ENS in the ecological wetland park, wherein predation was relatively easy, than in the rice paddy. However, the ENS of the gold-spotted pond frogs was highly probably overlapped with that of other competing frog species that shared some of the food sources. Nevertheless, since the stable isotope analysis showed that gold-spotted pond frogs fed on more diverse prey than their competitors, it would remain relatively easy to procure alternative food sources, which are less affected by the competition in an environment with abundant food. Therefore, for stable settling of gold-spotted pond frogs into habitats and preserving their population, establishing habitat environments with highly diverse food sources is crucial, following consideration of their feeding behavior.

**Abstract:**

The gold-spotted pond frog (*Pelophylax chosenicus*) is an endangered amphibian species in South Korea. In order to obtain ecological information regarding the gold-spotted pond frog’s habitat environment and biological interactions, we applied stable isotope analysis to quantify the ecological niche space (ENS) of frogs including black-spotted pond frogs (*P. nigromaculatus*) and bullfrogs (*Lithobates catesbeianus*) within the food web of two different habitats—an ecological wetland park and a rice paddy. The gold-spotted pond frog population exhibited a broader ENS in the ecological wetland park than in the rice paddy. According to the carbon stable isotope ratios, gold-spotted pond frogs mainly fed on insects, regardless of habitat type. However, the results comparing the range of both carbon and nitrogen stable isotopes showed that gold-spotted pond frogs living in the rice paddy showed limited feeding behavior, while those living in the ecological wetland park fed on various food sources located in more varied trophic positions. Although the ENS of the gold-spotted pond frog was generally less likely to be overlapped by that of other frog species, it was predicted to overlap with a high probability of 87.3% in the ecological wetland park. Nevertheless, gold-spotted pond frogs in the ecological wetland park were not significantly affected by the prey competition with competitive species by feeding on other prey for which other species’ preference was low. Since these results show that a habitats’ food diversity has an effect on securing the ENS of gold-spotted pond frogs and prey competition, we recommend that the establishment of a food environment that considers the feeding behavior of gold-spotted pond frogs is important for the sustainable preservation of gold-spotted pond frogs and their settlement in alternative habitats.

## 1. Introduction

The niche width of a species in an ecosystem represents species-specific characteristics, such as feeding habits, ecological characteristics, food preference and habitat selection; therefore, understanding the niche width from a population or community perspective can facilitate assessments of food web structure and function [1]. The niche concept was defined by Hutchinson [2] as an *n*-dimensional hypervolume has facilitated numerous community structure studies within ecosystems, in addition to the development of mathematical techniques for quantifying niche dimensionality, spatiotemporal change, trophic contribution, and inter- and intra-specific trophic diversity [1,3,4,5]. Such studies have enabled quantitative assessments of not only changes in community characteristics (e.g., species abundance, richness and evenness) that depend on the presence or absence of competitors and predators, but also of biological interactions based on positions within food webs, enhancing our understanding of organisms’ functions within ecosystem food webs [6,7].

Current understanding of niche width estimation is based on the heterogeneity of the feeding behavior of target organisms and ecological indices (e.g., richness and evenness) [1]. To consider the influence of available food sources, niche width analyses are supplemented with food type and source diversity data via stomach content and excrement analyses, providing accurate evidence on the foods consumed; However, under- or overestimation of the relative abundance of different food sources may occur due to limitations such as the detection of only specific foods that are undigested after being consumed within a short period of time.

Stable isotopes have been used as alternative tools for analyzing food sources and overcoming the limitations of existing methods. These isotopes can accurately evaluate trophic structure based on the fractionation effect exhibited by each isotope during food assimilation into the tissues of consumers in the upper trophic levels [7,8]. The carbon (C) stable isotope ratio (δ^13^ C) provides insights into the food sources of a consumer, as the fixed value does not change significantly from photosynthesis through feeding, digestion, and assimilation, with a difference lower than 1‰ between prey and predator. In contrast, nitrogen (N) stable isotope ratios (δ^15^ N) are typically enriched in consumers relative to the concentrations in the diets, and the concentrations in predators can be up to 3.4‰ higher than those in prey [1,9]. A bi-plot consisting of horizontal and vertical axes of such standardized isotopes can be used to determine the isotopic niche of a target organism for the prediction of population characteristics such as prey diversity and feeding interactions between organisms, in addition to their internal variability [3,4,10,11]. One approach for analyzing isotopic niche width is the comparison of stable isotope ratio variance within a population or between communities [1].

δ^13^ C–δ^15^ N stable isotope bi-plots are useful for identifying the isotopic niches of organisms within a food web and for the quantitative interpretation of community structure based on niche characteristics (range of δ^13^ C and δ^15^ N, etc.). However, stable isotope bi-plots have some limitations for assessing the impacts of individual species on trophic diversity and redundancy within communities. Layman et al. [3] proposed the concept of community-wide metrics, which quantifies isotopic niches of individual species and their relative positions, to reveal specific characteristics of a community structure within a food web. The total area of the convex hull (TA) in a δ^13^ C–δ^15^ N bi-plot and five metrics derived from it, δ^15^ Nrange (NR), δ^13^ C range, mean distance to centroid (CD), mean nearest neighbor distance (NND), and standard deviation of the NND (SDNND), reveals the responses of a population to environmental changes in the form of numerical values, which can be used not only to compare populations within a community, but also to analyze spatiotemporal changes in the overall community structure. However, community-wide metrics are estimates that do not consider uncertainties that may occur during sampling and population mean calculations, whereas TA, which forms the basis of community-wide metrics, is sensitive to sample size, presenting challenges when comparing populations (or communities) with different sample sizes. To address this problem, multivariate ellipse-based metrics, based on Bayesian inference techniques, that incorporate standard error in the estimates are used. Here, as standard ellipse area (SEA) can be derived by considering the sample size within a community (in this case, SEAc), communities with different sample sizes can be directly compared [4]. Such statistical methods have been applied to analyze the isotopic niches of various taxonomic groups both at the individual level, such as in benthic macroinvertebrates, amphibians, fish, and birds, and at the community level, such as in zooplankton [7,10,11,12,13,14].

All amphibians feed heavily on invertebrates; thus, their impact on this food source can be prodigious. In many habitats, amphibians are a major ‘conveyor belt’ that provides the transfer of invertebrate energy sources to predatory animals higher up the food chain. Amphibians have an important role in ecosystem energy flow and nutrient cycling in many ecosystems [15]. Gold-spotted pond frogs (*Pelophylax chosenicus*) (Class: Amphibia, Order: Salientia and Family: Ranidae) have decreased in number in South Korea because of habitat destruction and the introduction of bullfrogs, an invasive species [16]. The gold-spotted pond frog has been listed under the Korean endangered wild animals and plants list (Class II) and in the International Union for Conservation of Nature (IUCN) Red List of Vulnerable Species [17]. Gold-spotted pond frogs, which inhabit wetlands, are observed in high frequencies in manmade environments, such as rice paddies, where problems now exist with agrochemical exposure and a continuous reduction of rice paddy areas [17,18]. In addition, compared to the prey preferences of black-spotted pond frogs, gold-spotted pond frogs have a high preference for active prey that respond quickly to environmental changes in the habitat, making them more likely to react sensitively to habitat alteration that induces diet changes [19].

Recently, programs to establish alternative habitats for conserving gold-spotted pond frog populations have been pursued [20,21,22]. Although such alternative habitats offer additional ecosystem services in the form of species diversity and ecological parks and research avenues, understanding their habitat characteristics and interactions with other species would facilitate the implementation of effective strategies for guaranteeing stable communities in such alternative habitats [23,24]. However, most community restoration and conservation studies have focused on the physicochemical characteristics of the habitat and the behavioral ecology (e.g., home range size and movement distance) of single gold-spotted pond frogs within habitats. Few studies have explored population stability based on community structure and function perspectives within alternative ecosystems for this species [16,24,25,26,27]. Insights on community structure, including competition and predation interactions among gold-spotted pond frogs and other species within alternative habitats, can be obtained based on basic biological data, while food web-based studies may be required to consider functional aspects such as food selectivity and interspecific interactions.

In this study, we used stable isotope ratios to measure the niche widths of gold-spotted pond frogs and other frog species, with which they have competitive and predatory interactions, using frogs sampled from different habitats. The aim of the study was to investigate how different environments influence their feeding behavior and positions within the food web. Community-wide metrics were used to compare community characteristics, while niche overlap probability values were used to quantitatively assess how biological interactions with other frog species in different environments influenced gold-spotted pond frog trophic diversity. The results of this study provide basic ecological data on habitat environments and interspecific competition, which may facilitate the establishment and maintenance of gold-spotted pond frog populations in alternative habitats in conservation programs.

## 2. Materials and Methods

### 2.1. Study Sites

Gold-spotted pond frogs are typically found in wetland environments. Therefore, when alternative habitats are created to protect their populations, the establishment of artificial habitats such as ecological parks is often considered [27]. We selected Gyeongancheon Ecological Wetland Park in Gwangju-si, Gyeonggi-do (37°27′26.91″ N, 127°18′15.22″ E) and rice paddies on opposite sides of Hali Bridge on the Mankyeong River in Wanju-gun, Jeollabuk-do (35° 53′ 26.60″ N, 127° 05′ 55.92″ E) as the study sites, which were designated as ecological park and rice paddy habitats, respectively (Figure 1).

In the ecological wetland park habitat, frogs were sampled from emergent hydrophyte communities near five shallow ponds, which were established as gold-spotted pond frog habitats. The ecological park habitats have high plant cover with submerged, floating, and emergent hydrophytes, including artificially planted sacred lotus (*Nelumbo nucifera*), common reed (*Phragmites communis*) and cattail (*Typha angustifolia*), and each pond is connected to a small water channel to facilitate free movement of the organisms along the waterways. Frogs in the rice paddy habitat were sampled mostly from rice paddies and waterways on the banks of Mankyeong River. Common reed (*P. communis*) was the dominant plant, and other hydrophytes, such as wormwood (*Artemisia selengensis* f. *serratifolia*), water chestnut (*Trapa japonica*), and silver grass (*Miscanthus sacchariflorus*) were found close by. The rice paddy habitat is vulnerable to habitat fragmentation because of concrete channels and human interference, including the use of farming equipment during cultivation seasons (Table 1).

### 2.2. Sampling and Pretreatment for Stable Isotope Analysis

Gold-spotted pond frog sampling was performed immediately after the breeding season between mid-May and late-July, 2018. In ecological wetland park habitat, 12 gold-spotted pond frogs and five black-spotted pond frogs were sampled between July 30 and August 1, and in rice paddy habitat, seven gold-spotted pond frogs, six black-spotted pond frogs, and four bullfrogs were collected between August 13 and 14. The frogs were captured in the surrounding wetlands and grasslands which constituted the main habitat for the frogs. Considering the high nocturnal activity of amphibians, sampling took place during the day and night using amphibian-catching nets and stake nets. Black-spotted pond frogs and bullfrogs, which are competitors and predators of gold-spotted pond frogs, respectively, were sampled using similar techniques [19,28]. Because gold-spotted pond frogs are endangered species, the necessary permits were obtained from the Han River Basin Environmental Office and Jeonbuk Regional Environmental Management Office before sampling.

The captured frogs were ethically killed as painless as possible using MS-222 and then processed for analysis. Individuals sampled were transported to the Isotope Ecology and Environmental Science Lab, Hanyang University, under refrigerated conditions and subsequently freeze-stored (−23 °C) until stable isotope sample extraction. Appropriate amounts (1–2 g) of stable isotope samples were extracted from the muscles of each frog. Other tissues, such as bone, blood, and skin, were excluded to ensure accurate measurements. All samples were homogenized after freeze-drying (−85 °C) for at least 48 h, and then the samples divided into C and N stable isotope samples. The C stable isotope samples were treated with acid (HCl) to eliminate inorganic C [29]. The samples were also treated with a mixture of chloroform: methanol (2:1 *v*/*v*) to eliminate lipids to ensure that the C stable isotope ratio would accurately reflect food consumed recently [30,31]. No separate pretreatment was applied to the N stable isotope samples before measuring the stable isotope ratios.

### 2.3. Stable Isotope Analysis

C and N stable isotope ratios in the samples were analyzed using an elemental analyzer-isotope ratio mass spectrometer (EA-IRMS, Isoprime, Cheadle, UK). Because the absolute quantities of C and N stable isotopes in the samples were very low, the isotope ratio was converted and expressed as a relative value compared to a reference standard. Vienna Pee-Dee Belemnite and air were used as the reference standards for C and N stable isotopes, respectively, and the abundances of the isotopes in the samples relative to in the reference standards were calculated as parts per thousand and expressed as δ^13^ C and δ^15^ N Appendix, Table A1), as follows:δ13Ctarget = ([13C/12C]target/[13C/12C]sample-1) × 1000 (‰)(1)
δ15Ntarget = ([15N/14N]target/[15N/14N]sample-1) × 1000 (‰).(2)

#### Application of Stable Isotope Ratios

The ENS, or isotopic niche widths, of the frog populations, including the gold-spotted pond frogs in the two habitats, were analyzed based on δ^13^ C–δ^15^ N bi-plots that were constructed using the measured C and N stable isotope ratios. ENS values of frog populations between habitats were compared to determine their positions within the food webs, and differences in food sources and trophic levels of the populations were compared according to the habitat environments. ENS was calculated using the Stable Isotope Bayesian Ellipses function (SIBER) in R v3.5.1 based on the obtained stable isotope ratios [4,32]. SIBER analysis calculates ENS based on the coordinates of the ellipses in the stable isotope ratio bi-plots, with TA and SEA calculated using a Bayesian statistics-based model [4,13]. In addition, because ENS considers the sample size of the groups being analyzed, SEAc could also be inferred and comparative analyses were possible while addressing the uncertainties associated with analyzing groups of different sample sizes. Further, the calculated ENS values were used to derive community-wide metrics, including NND, to assess the species ‘packing density’ within communities, indicating trophic redundancy, to compare community structure characteristics between the two habitats [3].

In order to evaluate the impacts of competitive and predatory interactions on the ENS of gold-spotted pond frog populations, the nicheROVER package in R was used to calculate the probability (%) of interspecies-niche overlap (under a user-defined probability alpha, α= 0.95). Niche overlap of species *A* with respect to species *B* was based on the overlapping area between the two species within the total niche of species *B*. Because interspecies-overlap has directionality, values measured in the reverse direction (niche overlap of species *B* with respect to species *A*) could also be derived. Niche overlap probability was calculated based on the degree of influence of the resource utilization capacities of the species, relative abundances within habitats, and C and N stable isotope ratios, enabling a comprehensive interpretation of the results [5].

## 3. Results and Discussion

The raw data of the C and N stable isotope ratios measured from the collected frogs specimen are summarized in Appendix A.

### 3.1. Comparison of ENS in Gold-Spotted Pond Frogs According to Habitat

The SEAc of the gold-spotted pond frogs from the ecological wetland park and rice paddy habitats were 15.31‰^2^ and 0.38‰^2^, respectively, indicating broader niches in the ecological wetland park habitat (Figure 2).

The stable isotope ratios of the gold-spotted pond frogs from the ecological wetland park habitat exhibited a wide distribution along the horizontal and vertical axes. The difference in N stable isotope ratios between trophic levels is generally 3.4‰ [33]. Considering the range of the N stable isotope ratios of the gold-spotted pond frogs in the ecological wetland park habitat (3.50–15.12‰), there were differences at two or more levels, indicating that gold-spotted pond frogs exhibited omnivorous feeding habits. The C stable isotope ratios ranged from −29‰ to −24‰. In other wetlands on the Korean Peninsula, the C stable isotope ratio of aquatic insects (*Dytiscus marginalis*, Order Odonata; larvae) and terrestrial insects (Family Asilidae, Genus *Dolomedes* and Order Odonata; adult) range from −28‰ to −23‰, whereas those of zooplankton and aquatic plants were reported to range from −34‰ to −29‰, and −30‰ to −26‰, respectively [34]. In the present study, the aquatic insects captured from the ecological wetland habitat (*Enallagma cyathigerum*, *Ephemerella dentata*, *Neocaridina denticulate*, *Neverita didyma*, and *Paracercion calamorum*, Orders Odonata and Ephemeroptera) exhibited C stable isotope ratios ranging from −32‰ to −24‰ (unpublished data). When the C stable isotope ratio value ranges in the gold-spotted pond frog populations from the ecological park habitat were compared to those of the aquatic insects, the ranges were mostly similar, suggesting that these insects were the primary food sources of the gold-spotted pond frog populations in the wetland habitat.

In contrast, gold-spotted pond frog populations from the rice paddy habitat exhibited narrow C and N stable isotope ratio ranges, indicating that their food sources were much more limited than that of the gold-spotted pond frogs from the wetland habitats, and that the individuals were largely feeding on similar food sources. The C stable isotope ratios ranged from −24‰ to −23‰, indicating very levels of low prey diversity. Given that there was little difference in the N stable isotope ratios among individuals these frogs may have fed on prey of similar trophic positions.

Gold-spotted pond frogs are strongly omnivorous, feeding on either plant material or animals depending on feeding opportunities; therefore, their food sources vary depending on the environment [19]. Food sources shift based on food source availability within the habitat and its ease of capture [35]. Manmade ecological park habitats consist of various habitats that are maintained simultaneously, including ponds, grasslands, forests, and nearby rivers. Unlike rice paddies, ecological parks have forest sections that are positively correlated with insect biodiversity [36,37]. Moreover, ponds within ecological parks provide habitats for various aquatic insects, resulting in high trophic diversity for gold-spotted pond frog diets. Various aquatic plants covering the surface of ponds can provide a place to hide for many insects, and gold-spotted pond frogs uses them to feed [16]. In contrast, rice paddy habitats are relatively simple and show low trophic diversity. The observed range of feeding activity was low because of the relatively limited access to food sources resulting from low connectivity among rice paddy habitats and nearby waterways or land. Therefore, the differences in ENS of gold-spotted pond frogs between the ecological wetland park and the rice paddy habitats were attributable to the different environments in the habitats.

### 3.2. Impact of Biological Interactions: Comparison of ENS and Overlap Probability of Frog Communities in the Two Habitats

The SEAcs of black-spotted pond frogs, a competitor of gold-spotted pond frogs, in the ecological wetland park and rice paddy habitats were 1.08‰^2^ and 0.42‰^2^, respectively, indicating a similar ENS regardless of habitat (Figure 3A,B). These results further demonstrate that the ENS was much narrower for black-spotted pond frogs in the ecological park habitat than for gold-spotted pond frogs (Figure 3A). Both gold-spotted and black-spotted pond frogs have generalist feeding habits and respond to the movement activities of their food sources; however, in the same habitat environment, black-spotted pond frogs have a higher predation success probability for food sources that crawl on the ground and are commonly found on land. In contrast, old-spotted pond frogs prefer food sources that fly well and are active [19]. Recent behavioral studies of gold-spotted pond frogs reported that they engage in long-distance feeding activities over wide areas [24,27]. Therefore, gold-spotted pond frogs occupy a relatively wider ENS than black-spotted pond frogs when there is a great diversity of food sources and movement within the habitat is uninhibited.

In the ecological wetland park habitat, there was a significant difference in the C and N stable isotope ratios between gold-spotted pond frogs and black-spotted pond frogs, while in rice paddy habitat, the difference was relatively small (Figure 3). Some of both frogs them showed a difference in C stable isotope ratio of less than 1‰, suggesting that they fed on similar prey. However, other frogs fed on different types of prey with a difference of approximately 4–5‰. In the case of N stable isotope ratio, the trophic positions of prey fed by the two frog species were different (Figure 3A). In contrast, gold-spotted pond frogs and black-spotted pond frogs collected from the rice paddy not only showed a difference of less than 1‰ in C stable isotope ratios, but also showed a difference of less than 3.4‰ in the N stable isotope ratios, indicating both species fed on similar-origin prey that were positioned on the same trophic level (Figure 3B).

Bullfrogs, which were only captured from the rice paddy habitat, had an SEAc of 6.70‰^2^, indicating a wider ENS than that in the gold-spotted pond frogs (0.38‰^2^) and in the black-spotted pond frogs (0.42‰^2^) in the same habitat (Figure 3B). The niche space in the vertical direction appeared particularly wide, and comparison of the N stable isotope ratio values between individuals in the lower and upper niches revealed differences in trophic levels of one or more levels, indicating that the bullfrogs’ food sources ranged from low to high trophic levels. When compared with that of the gold-spotted and black-spotted pond frogs, bullfrogs with similar a distribution did not exhibit large differences in C stable isotope ratio values between individuals, indicating that they exploited similar food sources in the rice paddy habitat. In some bullfrog individuals, the N stable isotope ratio values were slightly higher than those of the other frog species; however, the differences were under 3.4‰^2^ and did not vary across trophic levels. Although bullfrogs are predators with strong omnivorous habits [38], a previous study reported that they generally feed on insect and Gastropoda food and predate on higher trophic levels, such as classes Amphibia, Aves, Mammalia, only transiently [28]. Only two of the four bullfrogs collected in the rice paddy habitat had higher N stable isotope ratios than that of the other frog species. However, the difference in N stable isotope ratio values among bullfrogs and other species in the same habitat was generally less than 3.4‰, suggesting that bullfrogs fed on prey of similar trophic positions.

Frog populations in the ecological wetland park and rice paddy habitats showed slight differences in C and N stable isotope ratio values (Figure 3A,B), which may be attributed to differences in food sources across the two habitats. Among community-wide metrics, Layman et al. [3] observed that nearest neighbor distance (NND) was relatively higher in the ecological park habitat food web (1.78) than in the rice paddy habitat food web (1.44, Figure 4). NND is primarily determined by variations in C isotope ratios in bi-plots; therefore, it can be applied in primary food resource diversity comparisons within a food web, and a decreased value can be interpreted as an increase in trophic redundancy between individuals [3,14]. When the food environments of the frog communities were compared within the ecological wetland park and rice paddy habitats, the results showed that food sources were more diverse in the ecological wetland park habitat than that in the rice paddy habitat. In addition, trophic redundancy between individuals was high within the rice paddy habitat, with relatively low levels of prey diversity. Considering the competition between gold-spotted pond frogs and other frog species in both habitats, competition is expected to be fiercer in the rice paddy habitat where food sources must be shared because of low trophic diversity.

Niche overlap trends among the frog populations in both habitats also varied due to the habitat environment. Black-spotted pond frogs, which competitively interact with gold-spotted pond frogs, showed a very high probability (87.3%) of overlap with the ENS of the gold-spotted pond frogs in the ecological wetland park habitat, whereas such probability was very low (9.06%) in the rice paddy habitat. In contrast, gold-spotted pond frogs exhibited low probability of niche overlap based on the ENS of black-spotted pond frog in the ecological park habitat (4.6%) and rice paddy habitat (9.41%) (Figure 5A; Figure 6A). Although niche overlap is influenced by the resource utilization capacity of a target species, relative abundance within habitats, and C and N stable isotope ratios [5], we could not measure the relative abundance of populations within each habitat. Thus, the relative abundance of each species within habitat was not considered when interpreting niche overlap probability. In the one-dimensional density plot of C stable isotope ratios of the gold-spotted and black-spotted pond frogs in the ecological wetland park habitat, the C stable isotope ratio range of black-spotted pond frogs overlapped partly with the range of the gold-spotted pond frogs, which demonstrated that the difference in food sources between species influenced the differences in the overlap probability values (Figure 5B).

Gold-spotted pond frogs have a high preference for active food sources which have a low probability of being consumed by black-spotted pond frogs [19]. Therefore, there is a low probability of gold-spotted pond frogs restricting the food sources which are available to black-spotted pond frogs in environments with abundant food sources; however, when food sources which are preferred by both species are abundant, there is a probability of larger amounts of such foods being consumed by black-spotted pond frogs, explaining why the ENS overlap probability of gold-spotted pond frogs by black-spotted pond frogs appeared high in ecological wetland park habitat [19]. In contrast, in the one-dimensional density plot of the C stable isotope ratios of the gold-spotted and black-spotted pond frogs in the rice paddy habitat, the isotope ratio ranges tended to be similar. However, only some parts overlapped in the one-dimensional density plot of N stable isotope ratios (Figure 6B). Based on these findings, the two species consumed food sources that had similar basal resources but different trophic positions in the rice paddy habitat and, as a result, exhibited low overlap probability. The probability of gold-spotted pond frogs overlapping the ENS of black-spotted pond frogs increased slightly, likely because of food source overlap following a decrease in food sources that were highly preferred by gold-spotted pond frogs in the rice paddy habitat, which had relatively low trophic diversity.

The above are some factors that may influence niche overlap. In addition, the influence of predators on other frog species and interspecific interactions should be considered, particularly in the case of the rice paddy habitats where bullfrog individuals were found. When ENS was compared between gold-spotted pond frogs and bullfrogs, a competitive relationship was inferred because the results did not reveal differences in trophic diversity (Figure 3B). The overlap probability of bullfrogs in the rice paddy habitat with respect to the ENS of gold-spotted and black-spotted pond frogs was low, at 0.84% and 0.61%, respectively, whereas gold-spotted and black-spotted pond frogs showed relatively high overlap probability with respect to the ENS of bullfrogs at 12.37% and 7.83%, respectively. These findings show that bullfrogs had very minimal direct influence on gold-spotted and black-spotted pond frog populations (Figure 6A). In the one-dimensional density plot of C stable isotope ratios, the stable isotope ratio values for the bullfrogs only minimally overlapped with those of the other species (Figure 6B). Although it was not possible to compare the overlap degree of the N stable isotope ratio with the other frog species due to a lack of signal (Figure 6B), direct comparisons of the N ratios between the bullfrogs and the gold-spotted pond frogs/black-spotted pond frogs showed a maximum difference of 2.9‰ and 4.42‰, respectively (Appendix A). Nevertheless, considering bullfrogs have a broader trophic diversity than that of the other frog species, the difference in the capacity to exploit available food sources may have influenced the observed overlap probability. This may be interpreted in a similar context as that of food selectivity in the gold-spotted pond frogs in the interactions between gold-spotted and black-spotted pond frogs, influencing the niche overlap probability of the two species. Moreover, because bullfrogs can consume foods that gold-spotted and black-spotted pond frogs cannot, there is less likelihood of restricting the feeding activities of the other frog species. Thus, niche overlap probability may have appeared as <1%. In contrast, bullfrogs primarily feed on insects, which is a typical food source for amphibians. Therefore, the niche overlap probability of bullfrogs appeared high due to the common feeding activities of the gold-spotted and black-spotted pond frogs. Particularly, gold-spotted pond frogs have a higher preference for mobile food sources, presenting a higher niche overlap probability with bullfrogs that tend to react and attempt to eat any objects that move around them [19,24].

Our results highlight the need for environments with high trophic diversity when selecting or establishing alternative habitats for gold-spotted pond frogs. Here, trophic diversity is not limited to one ecosystem but applies to both aquatic and terrestrial ecosystems. Therefore, it is essential to create an environment in which preferred food sources can be accessed freely from the two ecosystems, with no ecosystem fragmentation. Further studies are required to investigate essential food sources for gold-spotted pond frogs that would increase their competitiveness in biological interactions when establishing alternative habitats. In addition, gold-spotted pond frogs exhibited a tendency to consume foods with relatively high N stable isotope ratio values as they grew, representing higher trophic diversity, whereas there was no correlation between growth and N stable isotope ratio values in the black-spotted pond frogs in a previous study [18]. Such species-specific habits further supplement our results. In addition, bullfrogs, which were examined mostly based on their competitive relationship, influence amphibian communities, including gold-spotted pond frog communities, in various ways, such as through predation, breeding interference and disease transmission [39]. Therefore, it is necessary to investigate the influence of a greater variety of biological interactions based on the findings of the present study. Frogs spend time as tadpoles before metamorphosis into adults. Conducting similar analyses on gold-spotted pond tadpoles and their competitors and predators may facilitate sustained population conservation rather than during transient settlement only when the alternative habitat is established. Such additional studies can enhance our understanding of the feeding habits of gold-spotted pond frog populations and their degrees of interactions with other species. Findings from such studies would further enhance the establishment of stable alternative habitats for the conservation of gold-spotted pond frogs.

## 4. Conclusions

We used stable isotope ratios to analyze the ENS of gold-spotted pond frogs in two habitats to determine the influence of these habitats on feeding habits. The findings were applied to evaluate the influence of biological interactions based on comparisons with similar species. Measured C stable isotope ratios indicated that the sampled gold-spotted pond frog individuals fed on insects as their primary food source, regardless of the habitat, whereas N stable isotope ratios revealed slightly different distributions regarding trophic diversity depending on the habitat. Based on the habitat characteristics and stable isotope ratios, gold-spotted pond frogs in the ecological wetland park habitat with relatively high food abundance exhibited wider vertical ENS distribution, whereas those in the rice paddy habitat with somewhat limited food availability and individual mobility within the habitat exhibited narrow ENS with densely aggregated stable isotope ratio values.

In conclusion, habitat environments with broad food source varieties were more favorable for securing ENS in gold-spotted pond frogs. The degree of niche overlap with competitive species was greater in the ecological park wetland habitat. Black-spotted pond frogs were sharing the ENS of gold-spotted pond frogs, and the degree of overlap with competitive species may have increased with the increasing diversity of food sources available to gold-spotted pond frogs within the ecological park habitat. Nevertheless, concurrently, the impact of niche overlap does not seem to have played a large role as gold-spotted pond frogs could also secure competition-free food sources. In contrast, in the case of the rice paddy habitat, where gold-spotted pond frogs and its competitors were estimated to exhibit limited feeding activity through narrow ENS, the competition among species may have been be intensified through continuous deterioration of the food environment, despite the probability of the calculated niche overlap being lower than that of the ecological wetland park habitat. In addition, bullfrogs in the rice paddy habitat exhibited greater competition than predation to the gold-spotted pond frogs; however, the overlap degree of not only the ENS but also of the C stable isotope ratio range indicated that food diversity was low, meaning that the bullfrogs did not strongly influence the feeding activities of the gold-spotted pond frog populations. Unfortunately, we used a relatively small number of samples for ENS analysis and could not directly investigate the available food sources in the habitat and the prey fed on by the frog species. Considering factors such as food preference analyses in addition to ecological characteristics (population density and distribution trends of frog species) within a habitat using quantitative data (stable isotope ratios in food sources) and their impacts on competition and feeding environments obtained from further study to supplement these limitations would reveal a more comprehensive picture of the biological and environmental factors present, and facilitate the establishment of more appropriate alternative habitats for conserving endangered species such as gold-spotted pond frogs.

## Figures and Tables

**Figure 1 animals-11-00967-f001:**
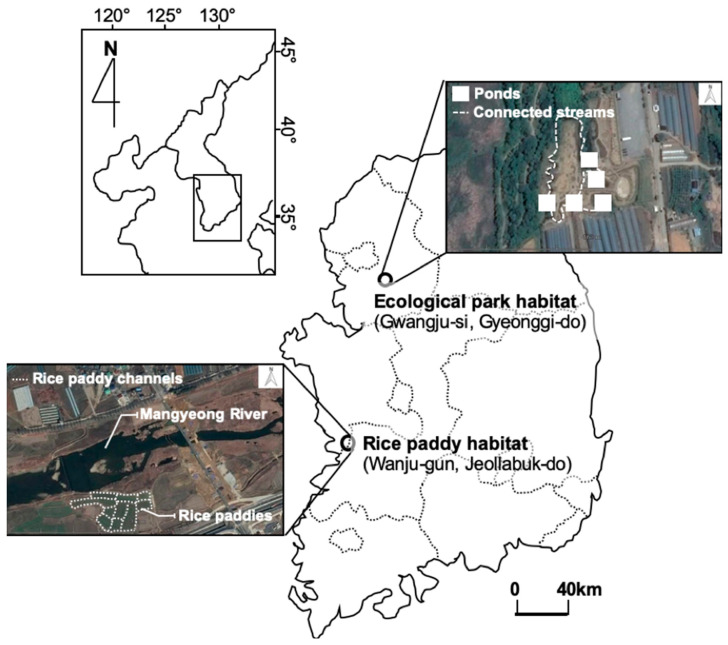
Study sites.

**Figure 2 animals-11-00967-f002:**
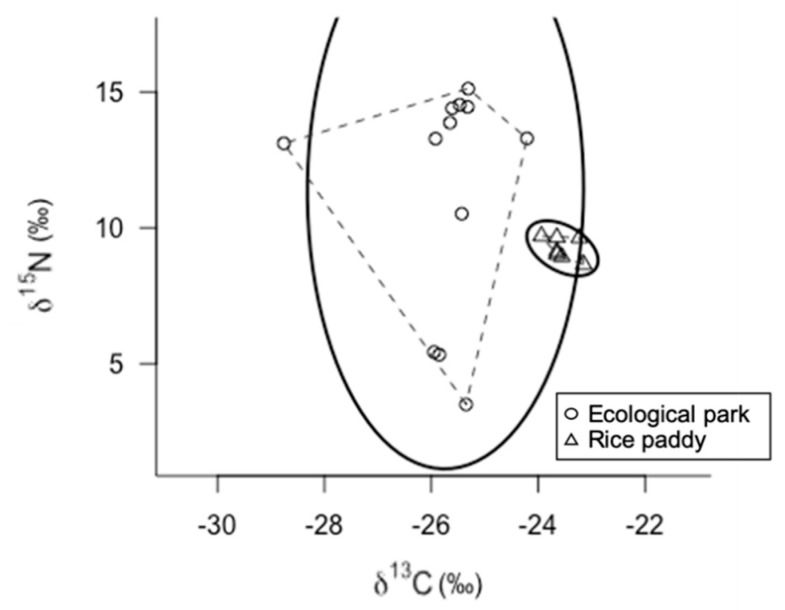
δ^13^ C-δ^15^ N bi-plot of gold-spotted pond frogs (*Pelophylax chosenicus*) captured from an ecological park wetland habitat (circles) and a rice paddy habitat (triangles); for each habitat, dashed lines indicate the convex hull areas (TA) and solid lines indicate the ellipse areas with *p* = 0.95 (*n* = 100).

**Figure 3 animals-11-00967-f003:**
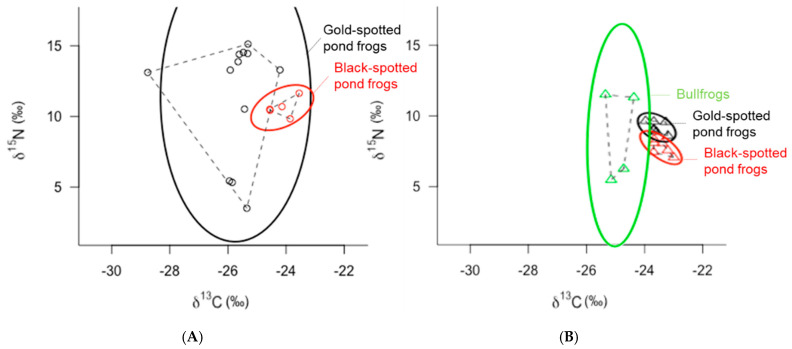
δ^13^ C-δ^15^ N bi-plot of gold-spotted pond frogs (*Pelophylax chosenicus*; black), black-spotted pond frogs (*P. nigromaculatus*; red), and bullfrogs (*Lithobates catesbeianus*; light green) captured from (**A**) an ecological wetland park habitat (circles) and (**B**) **a** rice paddy habitat (triangles): dashed lines indicate the convex hull areas (TA) and solid lines indicate the ellipse areas with *p* = 0.95 (*n* = 100).

**Figure 4 animals-11-00967-f004:**
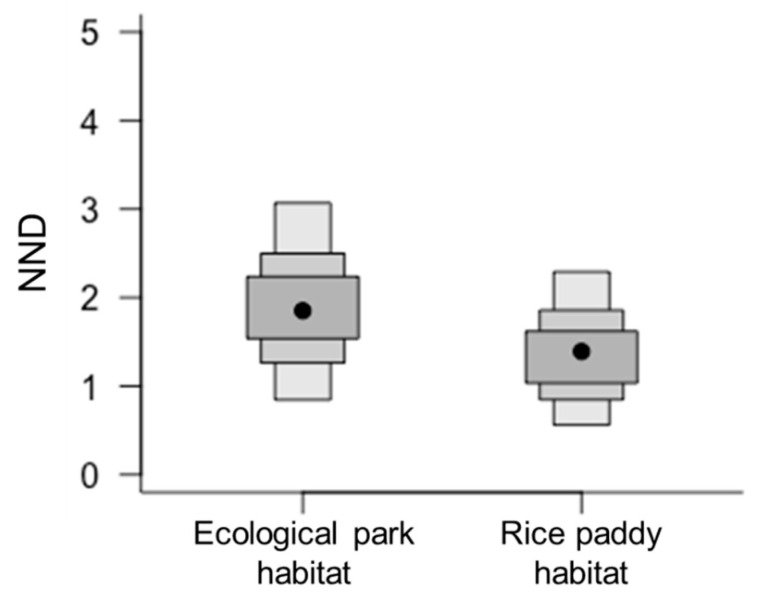
Comparison of community-wide metrics (mean nearest neighbor distance; NND) between ecological wetland park and rice paddy habitats, based on the posterior distribution of the means of each community’s carbon and nitrogen stable isotope ratios; mean NND: 1.78 in ecological wetland park habitat, 1.44 in rice paddy habitat.

**Figure 5 animals-11-00967-f005:**
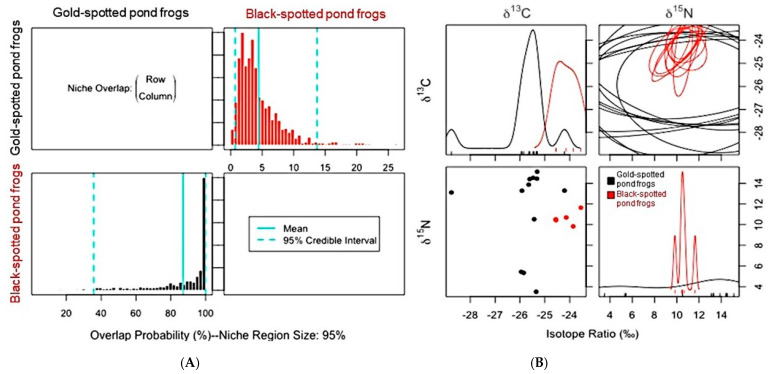
(**A**) Posterior distribution of the probabilistic niche overlap metric (%), niche region size (α) = 0.95 indicating that probability of one species (row) overlapped onto the niche of another species (column) between gold-spotted pond frogs (*Pelophylax chosenicus*) and black-spotted pond frogs (*P. nigromaculatus*) in the ecological wetland park habitat. (**B**) One-dimensional density plots of each carbon and nitrogen stable isotope; top-left and bottom-right, respectively, elliptical projections of the isotopic niche regions for each species and pair of stable isotopes; top-right, and two-dimensional scatterplots; bottom-left.

**Figure 6 animals-11-00967-f006:**
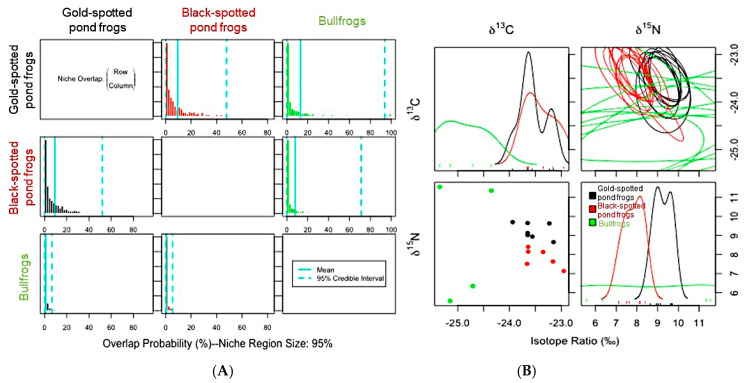
(**A**) Posterior distribution of the probabilistic niche overlap metric (%), niche region size (α) = 0.95 indicating the probability of one species (row) overlapping onto the niche of another species (column) among gold-spotted pond frogs (*Pelophylax*
*chosenicus*), black-spotted pond frogs (*P. nigromaculatus*). and bullfrogs (*Lithobates catesbeianus*) in a rice paddy habitat. (**B**) One-dimensional density plots of each carbon and nitrogen stable isotope; top-left and bottom-right, respectively, elliptical projections of isotopic niche regions for each species and pair of stable isotopes; top-right, and two-dimensional scatterplots; bottom-left.

**Table 1 animals-11-00967-t001:** Summary of the environmental characteristics of the ecological park habitat (Gwangju-si) and the rice paddy habitat (Wanju-gun).

	Habitat Types	Ecological Park	Rice Paddy
Characteristics			
Area (m^2^)		~9963 m^2^	~24,133 m^2^
Water body Characteristics (depth)		Ponds (<1 m) Connected streams (<0.5 m) Surrounding wetlands	Rice paddy fields (<0.1 m) Rice paddy channels (<0.3 m)
Surrounding Catchment Characteristics		Ecological park and parking lot Forest Agricultural area (vinyl house)	Rice paddies Bare patch Motorway
Aquatic Plants		Pond littoral and connected streams: *Alisma plantago-aquatica* Green algae *Myriophyllum spicatum* *Nymphoides indica* *Potamogeton crispus, P. distinctus, Spirodela polyrhiza*	Rice paddy fields: *Ceratophyllum demersum* *Oenanteh javanica* *Potamogeton distinctus* *Sagittaria trifolia* *Vallisneria asiatica* Rice paddy channels: *Potamogeton crispus, P. pusillus*
Benthic Macroinvertebrates		Family Atyidae; *Neocaridina denticulate,* Family Coenagrionidae; *Paracercion calamorum,* *Enallagma cyathigerum,* Family Ephemerelloidae; *Ephemerella dentata,* Family Naticidae; *Neverita didyma*	Family Atyidae; *Neocaridina denticulate,* Family Baetidae, Family Chironomidae, Family Culicidae, Family Naticidae; *Neverita didyma*

## Data Availability

The data presented in this study are available on request from the corresponding author.

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
