# Peer review of "Trophic Ecology of Endangered Gold-Spotted Pond Frog in Ecological Wetland Park and Rice Paddy Habitats"

_animals, 2021, doi:10.3390/ani11040967_

Round 1

Reviewer 1 Report

This study used stable isotope ratios to examine the trophic niche space occupied by the endangered gold-spotted pond frog in two different habitat types. They also examined the trophic niche space of two co-occurring frog species (black-spotted pond frog and bullfrog).

I don’t have any issues with the methods used in this study. It is a descriptive study, but has some interesting results. In particular, the fact that gold-spotted pond frogs have a much large niche space in the ecological park than in the rice paddy is interesting, as is the fact that bullfrogs have a much larger niche space than either of the native frogs. My only complaint is that some of the statements describing the results in the text don’t seem to agree with the data presented in the figures (see below). I suggest that the authors think hard about which aspects of their results they view to be most important and then convey those to the readers more clearly.

Line 45: Change “from” to “in”.

Lines 48-49: Reverse edit back to “information regarding the gold-spotted pond frog’s habitat environment”.

Lines 56-57: The mean N ratio of gold-spotted pond frogs was roughly the same regardless of habitat type. Thus, this seems like an inaccurate way to describe the results. The difference between habitats is really about the diversity of trophic levels at which they feed, not that they are primary consumers in one habitat and secondary consumers in another.

Lines 58-59: Delete “showing a slight increase in overlap”.

Line 59: Delete “in-/direct”

Line 60: Should be “frog” instead of “frogs”.

Line 61: Niche overlap with other frog species was fairly small in both environments, indicating that competition (at least for food resources) is unlikely to be a limiting factor in gold-spotted frog persistence. Thus, highlighting competition in the concluding sentence of the abstract seems to be an inaccurate summary.

Line 78: I am not sure what functional approaches are being referenced here.

Lines 86-89: I’m not sure what this sentence is trying to say. In what way are turnover times of foods consumed by upper trophic levels not reflected in gut contents?

Line 123: Replace “data sets” with “sample sizes”.

Line 128: Delete “because of their feeding habits”.

Line 129: Delete “going”.

Lines 128-129: Some amphibians alternate between terrestrial and aquatic environments, but many do not.

Line 131: Aren’t there easier ways to tell if wetlands are lost (e.g., direct observation) than by tracking whether amphibians are declining?

Line 132: Class is “Amphibia” not “Amphibians”.

Line 133: No need to give the genus since you just gave the scientific name.

Lines 137-139: Why does inhabiting rice paddies make gold-spotted pond frogs vulnerable to habitat conversion? The fact that they can persist in rice paddies as well as native habitat should make them less vulnerable.

Lines 157-159: Why does knowledge of interspecific interactions require a food web study? Many interspecific interactions are non-trophic and/or can be observed directly.

Lines 172-174: Why is an ecological park considered to be equivalent to a wetland?

Line 183: Delete “in Gyeongancheon Ecological Wetland Park”.

Line 189: Delete “near vegetation on the river bank”.

Line 190: Replace “whereas” with “and”.

Line 191: By “the area” do you mean rice paddies?

Line 192: Delete “also”.

Lines 233-234: Do you mean “Bayesian ellipses in the SIBER package of R v3.5.1”?

Lines 248-252: Neither of these sentences makes sense to me. Isn’t niche overlap expressed as a percentage because it is a proportion of the total niche space that is shared? In other words, it is a percentage of the total area rather than some probability of a given overlap? And isn’t niche overlap probability based on resource utilization capacities, relative abundances, and stable isotope ratios rather than being interpreted by them?

Results: I would begin the Results by giving the sample size – number of frogs from which stable isotope data were collected.

Line 259: Replace “was” with “is”.

Lines 270-272: The ellipses seem bigger than the convex hulls, but the stated areas are smaller.

Lines 282-283: You give the C ratios for the rice paddy but never state the N ratios.

Line 304: Replace “in” with “for”.

Lines 308-309: Change to “with minimal movement. Gold-spotted pond frogs, on the other hand, prefer food sources that”

Lines 324-326: You state that black-spotted pond frogs exhibit a similar range of C ratios as gold-spotted pond frogs, but based on Figure 3, this does not seem to be true.

Lines 326-328: You state that gold-spotted pond frogs have an N ratio of approximately 3.4% in the ecological park, but this does not seem to be true based on Figure 3.

Lines 329-330: You again state that C ratios in the ecological park are the same between gold-spotted and black-spotted pond frogs, but this does not appear to be true in Figure 3.

Lines 392-395: I have no idea what these two panels represent.

Lines 401-402: The proportion of gold-spotted pond frog niche space occupied by black-spotted pond frogs never appears to be high. This thus seems to be a false statement.

Line 407: Replace “diversity” with “positions”.

Lines 419-421: I again have no idea what these two panels represent.

Lines 434-435: The N ratio for bullfrogs actually seems to have high overlap with the other two species, so this seems to be inaccurate.

Lines 473-477: Gold-spotted pond frog C and N ratios in the rice paddy both appeared to be small subsets of those observed in the ecological park. Thus, I’m not sure why this distinction is being made between the two elements.

Line 478: I don’t recall food abundance ever being measured in this study in either habitat.

Line 484: Aren’t black-spotted pond frogs native to Korea? If so, why do you suggest they are invading the ENS of gold-spotted pond frogs? Don’t they just share niche space?

Lines 491-493: Change to “within a habitat using quantitative data (stable isotope ratios in food sources) and their insights into competition and feeding environments would reveal a more comprehensive picture of biological”.

Lines 494-495: Please be more specific. What conservation action would you recommend based on this study? Are frogs necessarily better off with a broader niche space? As long as they can find enough resources in the portion of niche space that they occupy, then won’t individual fitness be fine?

Reviewer 2 Report

This study examined niche space of some frog species in two habitats using stable isotope ratios. I think the authors tried to infer interspecific interactions and habitat suitability based on such information. However, the notion that quantification of niches space help understand competitive interspecific interactions is quite old, perhaps more than 40 years before. Current ecological theory indicates the notion is false, because a high niche overlap could indicate no severe competition under non-equilibrium conditions, and a low niche overlap could be the result of competitive interactions. Without other ecological information, such as vital rates or experimental manipulation of one species, little can be inferred of how the community is organized by competition. Stable isotope analysis is effective to know predator-prey interactions when candidate prey are specified, but it has only limited value for evaluating whether competition is important.  This study therefore has only descriptive value of frog diets, with little contributions to ecological mechanisms.

Reviewer 3 Report

The authors utilized C and N stable isotope analysis to test two major questions. 1) Does the Gold-Spotted Pond Frogs prey niche width differ between wetland parks and rice paddy habitats? And 2) Does the Gold-Spotted Pond frog prey niche width overlap with other frog species within wetland parks and rice paddy habitats.  The stable isotope approach is a relatively novel approach to determine if prey diversity differs between habitat types and if there is a potential for inter-specific competition within each habitat. Overall, the manuscript is well organized but does require revisions to support some of the manuscript claims. The manuscript documents  three main results:. Diet diversity is significantly reduced in rice paddy habitats relative wetland parks for the gold-spotted frog; there is increased prey niche overlap in rice paddy habitats, while the invasive bullfrog does not appear to be in direct competition with the native species. The manuscript can be shortened to emphasize these points, with caveats as suggested below. There are several places where the manuscript makes statements that are not backed up by either citations or data; some of these instances are noted below. The results are interesting and potentially important to managers who wish to improve habitat for the conservation of the gold-spotted pond frog.  Some minor and some more major issues (sample size, pseudoreplication, and the need for better methods reporting, especially with regards to euthanasia and animal ethics are necessary) are presented below.   

Simple Summary: It seems like the final conclusion is overly broad given the data presented. 

Abstract: Was habitat connectivity and food abundance also measured simultaneously?  If not, this is a broad statement without supporting data. Do they have less abundant food resources or less diverse food resources?

Introduction:

Lines 117-120 need citations. 

Lines 123-125: It does not look like the paper cited as “15” includes this golden spotted frog in its diet survey.  This paper compares two other Rana (now Pelophylax) species, but not the gold-spotted pond frog.  Because the paper is in Korean and neither it nor its English summary is readily available, it is difficult to determine whether the diet analysis cited in that paper also references the gold-spotted frog as indicated in the paper. 

Line 124 – It is unclear what the authors mean by “relatively higher food selectivity”

Lines 131-132: This sentence is redundant with the first in the paragraph.

Methods:

Line 161. Figure 1. What is the scale for the blown up images?

Line 177: What is meant by “Pre-treatment”?

Table 1 has line running through heading

Lines 181-182. The statement about frog predators and competitors needs a citation.

177-194: There is no statement of how the animals were euthanized and whether there was a process for humane treatment. 

Generally, the methods need more information.  How many nights were the frogs sampled? How many frogs were sampled each night for each species? Where all animals adults? Was there only wetland and one set of rice fields sampled (pseudoreplication), which would make the results very difficult to evaluate in a broader ecological context?

Results:

Line 240-241: Given that the wetland habitat may have more complex vegetative structure, could that explain the herbivorous component of the isotope signature?  Accidental vegetative ingestion while catching prey items may be more common in a more complex environment.

Lines 252-260: The methods do not mention any insect collection or surveys, yet the results state which species were present and their isotopic signatures as (unpublished data).  Why not include those surveys and the data within this paper? 

Lines 235: The SEAc values in do not match up with the values in the figure legend for Fig. 2 (lines 251).  If this is because one is SEA and the other is SEAc, please explain why the manuscript is presenting each one. 

Line 263: Again, it is difficult to determine whether this citation represents the gold-spotted frog as it is not available online, and the title does not mention this species, just a similar one. 

Line 270 – 272 – Is there data to support that aquatic plant cover facilitates patch connectivity and food availability?

Line 286 – 287 – Is there data to support that black spotted pond frogs have higher predation of insects which do not fly? Sentence should be revised.

Line 326 – 329 – What do the authors mean by more carnivorous? I am confused if they are eating higher up in the trophic food web why are their N ratios lower than the gold-spotted frogs?

Line 374 – What do the authors mean by move considerably? How can you classify prey that considerably moves versus a more stagnant species?

Conclusions:

In general, I do not disagree with the conclusions, but given that the manuscript does not explore or cite the differences in food resources available between the two sites, and given that they perhaps only sampled one rice field and one ecological wetland, the results need to be better qualified.

The discussion on the differences among frog species within each habitat (related to Fig. 3) needs to be shortened and qualified greatly based on the very small sample sizes for the Black-spottted pond frogs in the wetland and all species in the rice fields.  The same applies to the niche overlap arguments, and the probabilistic niche overlap metrics and region size. 

Fig. 3: the label for Black-spotted pond frogs has a misspelling (“fond”). 
